# No more hard prompts: SoftSRV prompting for synthetic data generation

## Abstract

We present a novel soft prompt based framework, SoftSRV, that leverages a frozen pre-trained large language model (LLM) to generate targeted synthetic text sequences. Given a sample from the target distribution, our proposed framework uses data-driven loss minimization to train a parameterized "contextual" soft prompt. This soft prompt is then used to steer the frozen LLM to generate synthetic sequences that are similar to the target distribution. We argue that Soft-SRV provides a practical improvement over common hard-prompting approaches that rely on human-curated prompt-templates, which can be idiosyncratic, labor-intensive to craft, and may need to be specialized per domain. We empirically evaluate SoftSRV and hard-prompting baselines by generating synthetic data to fine-tune a small Gemma model on three different domains (coding, math, reasoning). To stress the generality of SoftSRV, we perform these evaluations without any particular specialization of the framework to each domain. We find that Soft-SRV significantly improves upon hard-prompting baselines, generating data with superior fine-tuning performance and that better matches the target distribution according to the MAUVE similarity metric.

## 1 Introduction

In recent years, pre-trained large language models (LLMs) have proven to be effective in generating synthetic natural language training data (Gunasekar et al., 2023; Li et al., 2023; Eldan & Li, 2023; Mukherjee et al., 2023; Mitra et al., 2023; Abdin et al., 2024). This is particularly true when the synthetic data is used to pre-train or fine-tune smaller language models, enabling performances that rival models that are orders of magnitude larger (Liu et al., 2023). There are several motivations for generating and using synthetic training data; chief among them is the need to train models for domains where little natural high-quality text may be readily available or may be difficult to procure.

In order to generate synthetic text, a significant amount of human-driven prompt engineering is invested into developing prompts that steer the generating LLM into producing high-quality text from a targeted domain while also encouraging sufficient diversity. This point was very nicely summed up by the authors of the open-source synthetic text repository Cosmopedia (Ben Allal et al., 2024), when recounting their attempt to recreate a large synthetic dataset similar to the one generated to train Phi 1.5 (Li et al., 2023):

> "Heads up: If you are anticipating tales about deploying large-scale generation tasks across hundreds of H100 GPUs, in reality most of the time for Cosmopedia was spent on meticulous prompt engineering."     – Ben Allal et al. (2024)

Furthermore, and especially in the case of generating fine-tuning data for targeted domains (e.g., coding, math, customer service), this manual process may need to be repeated and refined per-domain, or even per sub-domain (e.g., per coding language, math subject, service department). Apart from the human engineering cost, these manual prompting approaches do not directly optimize a data-driven objective. Rather they depend on human-in-the-loop style feedback for manually adjusting the prompt templates, resulting in approaches that lack robust mechanisms for aligning the LLM's generated data with the desired distribution.

To address these issues, we propose an algorithmic framework, *Soft prompt-based Synthesis with Randomized Variation* (SoftSRV), that leverages *soft-prompting* (also known as prompt-tuning) for

synthetic text data generation. Soft prompt training is a parameter efficient tuning method and requires a relatively limited amount of compute (Lester et al., 2021; Li & Liang, 2021). Perhaps equally important, since the soft prompt is trained using a data-driven training algorithm, it requires essentially no human-in-the-loop intervention, enabling the process to be readily deployed across many different domains. Furthermore, soft prompts themselves can allow for more expressive input contexts to the generating LLM compared to natural language hard prompts. A soft "token", represented by a dense vector, is not restricted to correspond to a particular discrete natural language token (e.g., sub-word or character). This intuitive observation is formalized in Petrov et al. (2024b), which shows that in specific settings soft prompts can induce an LMM to produce an exponential (in sequence length) number of text completions, while hard prompts only allow for a linear number of completions.

Why do we expect soft-prompting to be effective for targeted synthetic generation? Prior theoretical research on fine-tuning language models suggests that a data-driven optimization of soft prompts guides a pre-trained model towards specific concepts or tasks it has already learned, essentially steering the model towards a relevant subspace of interest (Wies et al., 2023; Petrov et al., 2024b;a). Our goal is to use soft prompts to steer the pre-trained model towards generating text that most resembles the target distribution. Subsequently, fine-tuning a smaller model using the generated data provides an effective way to transfer knowledge from the larger model to the smaller model.

Unlike in typical prompt-tuning approaches, we do not prepend a soft prompt to an existing hard prompt, but instead use the soft prompt alone as input context to the LLM. We use a sample of text sequences (i.e., a sample from the desired target text distribution) and language-modeling loss to learn a parametrized soft prompt. Once trained, the soft prompt can be varied by conditioning on a context vector derived from an example sequence, allowing for additional expressive power and the potential to better fit different regions of a potentially complex target distribution.

Our contributions presented in this work are as follows:

- We demonstrate that soft prompts can be effectively trained for the purpose of targeted synthetic text generation used to fine-tune downstream models.

- We investigate the value of learning parameterized families of soft prompts that can be conditioned on an input context, finding they are critical for best fitting complex target distributions.

- Our empirical evaluations on coding, math, and reasoning tasks find superior downstream performance for models fine-tuned on SoftSRV generated text compared to that of models fine-tuned on data generated by baseline hard-prompting approaches.

- We show that the SoftSRV approach is general and offers greater versatility than hard-prompting approaches as it can be readily applied across different domains with minimal manual intervention.

- We measure the similarity of the generated data to the target distribution using the MAUVE metric and observe that SoftSRV methods align most closely with the target distribution.

## 2 PROPOSED APPROACH

In this section, we introduce the general SoftSRV framework as well as a few specific instantiations that are studied in this work. First, we start with some basic notation and terminology.

Given a vocabulary $\mathcal{V}$ of textual tokens, let $\{x_1, \ldots, x_n\}$ denote a sample of $n$ text sequences, belonging to the set of all possible sequences $S^m$ of a finite maximum length $m$, drawn according to a fixed but unknown distribution $\mathcal{D}$. Although we are not able to directly sample additional sequences from $\mathcal{D}$, our goal is to synthesize sequences that could have plausibly been drawn according to $\mathcal{D}$. We assume access to a (frozen) LLM, denoted $L : S^m \rightarrow S^m$, where we input and output sequences of equal fixed length $m$ for notational simplicity and without loss of generality. Furthermore, we explicitly decompose the LLM, $L = H \circ E$, where $E : S^m \rightarrow \mathbb{R}^{d \times m}$ represents the initial embedding layer that embeds each token of the input sequence to a $d$-dimensional dense vector, and $H : \mathbb{R}^{d \times m} \rightarrow S^m$ represent the remainder of the language model that maps the embedded tokens to the output sequence. In contrast to the prompt tuning methods of Lester et al. (2021) in the standard fine-tuning setting, we do not prepend the learned soft tokens to a hard prompt after it passes the

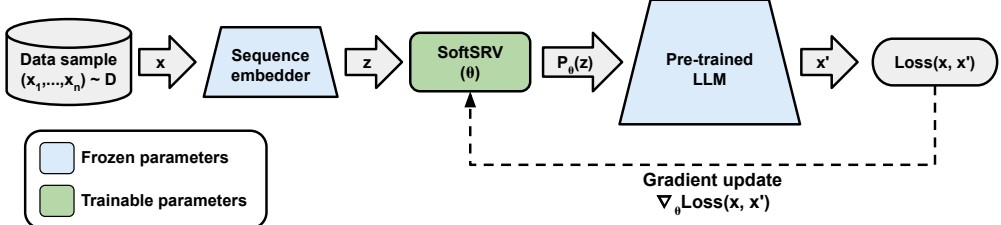

Figure 1: A diagram illustrating the training workflow of the SoftSRV framework. An example sequence $x$ is embedded into a dense vector $\mathbf{z}$ via a (frozen) sequence encoder model. The SoftSRV model, parameterized by $\theta$ and conditioned on the embedding $\mathbf{z}$, produces a soft prompt $\mathbf{P}_\theta(\mathbf{z})$. This is then fed to a (frozen) pre-trained LLM, which produces a synthetic example $x'$. Similar to autoencoder-based training, the gradient of a next-word-prediction "reconstruction" loss is computed and used to update the SoftSRV parameters.

initial embedding layer, $E$. Instead, for our approach we discard $E$ entirely method and rely on the frozen model $H$.

The SoftSRV framework seeks to synthesize sequences similar to those drawn from $\mathcal{D}$ by training a "soft prompt", i.e. a dense embedding (or parameterized family of embeddings) $\mathbf{P} \in \mathbb{R}^{d \times t}$ consisting of $0 < t < m$ "soft-tokens". A successfully trained soft prompt, $\mathbf{P}$, should generate a sequence $x = H(\mathbf{P})$, via frozen model $H$, that has a high likelihood of occurring under the distribution $\mathcal{D}$. More generally, we can sample several different sequences from a fixed prompt, $x, x', x'' \ldots \sim H(\mathbf{P})$, by using randomized temperature-based decoding. Although temperature sampling alone does allow for some variability, we can further increase the variety of generated text by using a *contextual* soft prompt, $\mathbf{P}(\cdot) : \mathbb{R}^{d_e} \to \mathbb{R}^{d \times t}$. A contextual soft prompt can be conditioned with different context vectors $\mathbf{z} \in \mathbb{R}^{d_e}$, during training and generation, to induce variations of the soft prompt.

Before introducing specific soft prompt instantiations, we describe the SoftSRV training procedure which is common throughout and also illustrated in Figure 1. In addition to the sample of data $(x_1, \ldots, x_n)$ and frozen LLM ($H$), we assume access to a sequence embedding function $\mathrm{emb}(\cdot) : S^m \to \mathbb{R}^{d_e}$ and we let $\theta$ denote the trainable parameters of the (contextual) soft prompt $\mathbf{P}_\theta(\cdot)$. During training, each training sequence is embedded $\mathbf{z}_i = \mathrm{emb}(x_i)$ and used to generate a conditioned soft prompt $\mathbf{P}_\theta(\mathbf{z})$, which is fed into the frozen LLM $H$ to produce a new sequence $x_i' \sim H(\mathbf{P}_\theta(\mathbf{z}_i))$ using the standard sequential (next token) generation. A standard causal (next-word) prediction loss, denoted $\ell(\cdot, \cdot)$, is backpropagated through the network up to the soft prompt layer $\mathbf{P}_\theta$, and an SGD-style update is applied to $\theta$ using the gradient $\nabla_\theta \ell(x_i, x_i')$. This loss can be thought of as a "reconstruction" error and the entire pipeline is akin to an auto-encoder. Viewing the pipeline through this lens, it is apparent that the sequence embedder $\mathrm{emb}(\cdot)$ should be sufficiently "lossy" in order to avoid making the learning problem trivial. This lossiness can be enforced by restricting the dimension $d_e$ of the embedding, for example.

Once the contextual soft prompt $\mathbf{P}(\mathbf{z})$ has been trained, we can then generate synthetic data by prompting the LLM using $\mathbf{P}(\mathbf{z})$ as embedded input context for different choices of context vector $\mathbf{z}$. A natural choice is to sample embeddings $(\mathbf{z}_1, \ldots, \mathbf{z}_n)$ derived from the data sample set $(x_1, \ldots, x_n)$. We now introduce a few specific SoftSRV parameterizations studied in this work.

## 2.1 Non-contextual soft prompt ($SS_{NSP}$)

The simplest parameterization treats the $dt$ entries of a soft prompt, $\mathbf{P} \in \mathbb{R}^{d \times t}$, directly as trainable parameters, i.e., $\theta = \mathbf{P}$, resulting in the following objective:

$$\mathrm{argmin}_\theta \sum_{i=1}^{n} \ell(H(\mathbf{P}), x_i) , \tag{1}$$

where it is understood that, in practice, the argmin over $\theta$ is only approximated via SGD. This parameterization is an instance of a *non-contextual* soft prompt, i.e., any context $\mathbf{z}$ is ignored. Despite the

lack of context, the synthesized output may still be diversified by using non-greedy (i.e., temperature sampling) decoding during LLM generation.

## 2.2 MIXTURE OF PROMPTS ($SS_{MPk}$)

Here, we train $k$ "basis" soft prompt matrices and define the final soft prompt, as a mixture of these bases. More precisely, in this variant the parameter set is $\theta = \{\mathbf{P}_1, \ldots, \mathbf{P}_k, \phi\}$, where $\mathbf{P}_i \in \mathbb{R}^{d \times t}$ are the basis prompts,

$$\mathbf{P}_\theta(\mathbf{z}) = \sum_{i=1}^{k} w_i \mathbf{P}_i, \quad (w_1, \ldots, w_k) = W_\phi(\mathbf{z}), \tag{2}$$

and $W_\phi(\cdot) : \mathbb{R}^{d_e} \to \mathbb{R}^k$ is a learned softmax function with parameters $\phi \in \mathbb{R}^{d_w}$. The trained $SS_{MPk}$ prompt is then the SGD solution to $\mathrm{argmin}_\theta \sum_{i=1}^{n} \ell(H(\mathbf{P}_\theta(\mathrm{emb}(x_i)), x_i)$.

The intuition behind this formulation is for each learned basis soft prompt $\mathbf{P}_i$ to encode a different aspect (mode) of the target data distribution and have each training example $x_i$ approximated by a mixture of these modes (similar to the intuition behind mixture or topic models (Hand, 2018)). Previous prompt-tuning works have also made use of a mixture of soft prompts, albeit not focused on training data synthesis (Qin & Eisner, 2021; Dun et al., 2023).

## 2.3 MLP CONCATENATED ($SS_{MC}$)

We consider a collection of $t$ small MLPs, whose output is concatenated to generate the final soft prompt. Let $F_{\phi_i} : \mathbb{R}^{d_e} \to \mathbb{R}^t$ denote the $i$th MLP with parameters $\phi_i$, and $\theta = \{\phi_1, \ldots, \phi_t\}$ denote the parameters for the collection of MLPs. Then, we define:

$$\mathbf{P}_\theta(\mathbf{z}) = \left[ F_{\phi_1}(\mathbf{z}), \ldots, F_{\phi_t}(\mathbf{z}) \right], \tag{3}$$

and the trained $SS_{MC}$ soft prompt is the SGD solution to $\mathrm{argmin}_\theta \sum_{i=1}^{n} \ell(H(\mathbf{P}_\theta(\mathrm{emb}(x_i)), x_i)$.

This parameterization is the most expressive that we consider, in that each "soft-token" in the soft prompt is computed using a distinct non-linear transformation of the context vector $\mathbf{z}$.

## 3 EMPIRICAL EVALUATION

In the our empirical evaluation of the SoftSRV framework, we consider a supervised fine-tuning setting where a small Gemma 2B model (Team et al., 2024) is fine-tuned on synthetic data generated by a larger decoder-only language model across several different benchmark datasets.

### 3.1 DOMAINS AND DATASETS

In order to demonstrate the generality of the proposed approach, we consider fine-tuning for several disparate domains (coding, mathematics, reasoning) using the same exact pipeline with no particular specialization to any of the particular domains. We briefly describe the specific benchmark we use from each domain.

**Code – MBPP (Austin et al., 2021).** For the coding domain, we consider the "Mostly Basic Python Problems" (MBPP) benchmark, which consists of short Python programming exercises, e.g. "Write a python function to find the first repeated character in a given string" and answers written in Python code. The task is evaluated using a 3-shot prompt (i.e. a prompt pre-pended with three instructional examples) and pass@1 metric, measuring if a single top generated result is correct.

**Math – GSM8K (Cobbe et al., 2021).** For the math domain, we use the grade school math word problems from the GSM8K benchmark. This dataset contains only highly-curated word problems written by humans that are conceptually simple, but require multi-step reasoning. For this generative task we use a 5-shot prompt, again, measuring the pass@1 metric.

**Reasoning – BoolQ (Clark et al., 2019).** For the general reasoning domain, we consider the binary question answering dataset from the BoolQ benchmark. The questions arise organically from

anonymized Google search queries which can be answered as either 'true' or 'false'. Each question and answer is paired with a passage (average length of 108 tokens) that is extracted from a relevant Wikipedia page. This is evaluated as a scoring/classification task and accuracy is reported.

The above chosen benchmarks aim to cover a wide variety of tasks, each with varying degrees of complexity – both in terms of solving the task and in terms of generating synthetic data for the task. The MBBP task requires basic Python programming knowledge to solve, but the questions are generally short, follow a similar pattern and are concerned with a relatively narrow set of themes. The GSM8K task requires basic math and language comprehension skills, but generating the problem is arguably even harder than solving it. It requires generating a premise containing several numerical quantities and then a question that can be answered using the provided information in a non-trivial fashion. Nonetheless the premises are somewhat formulaic and thematically similar. The BoolQ reasoning task, which requires general reading comprehension to solve, is perhaps the most difficult task to generate synthetic data for. Generating a problem requires writing a long (relative to MBPP and GSM8K) passage on an arbitrary topic that contains a collection of facts, but that does not necessarily stick to any formula or theme, and then generate a true/false question that can be answered directly by the passage. As we shall see in the empirical evaluation that follows, the level of difficulty in generating a high-quality question can impact the relative quality and value of generated synthetic data.

## 3.2 HARD-PROMPTING BASELINES

Typical hard prompt engineering approaches involve manually creating prompt templates that are then seeded with text from the desired target domain, typically taken from the training set. To give a (simplistic) illustrative example, a template could be:

```
Consider the following [article], write a textbook quality
summary of the topic suitable for a high-school audience,
```

where the placeholder `[article]` would be replaced with example texts from training fold, producing several distinct prompts. In this study, we consider the following two hard-prompting variants.

The first, denoted simply as hard prompt (HP), uses a domain-specific hard prompt template to generate a question followed by another domain specific hard prompt template to generate answers (the detailed workflow is discussed in Subsection 3.3). In Appendix B, we provide the exact templates used by the HP method. To reach these templates, we undertook several iterations of hard prompt engineering and reported the result of the best performing method. In particular, we found that prompting for a "diverse" set of questions was crucial (a comparison plot is presented in Appendix A.5).

The second approach, hard-prompting with self-refinement (HP$_{SR}$), similarly uses a hard prompt template to generate questions but also iteratively conducts several rounds of self-critique to improve or accept the question (Madaan et al., 2023). Again, critique and refinement prompts are in Appendix B.

## 3.3 EMPIRICAL EVALUATION PROCEDURE

For each benchmark dataset and each data generation approach, the evaluation pipeline is as follows.

**1. Train SoftSRV prompts.** For the SoftSRV methods, we first train the soft prompt parameters with a frozen large decoder-only LM backbone using the questions found in the training fold of the dataset, which serves as our sample from the target distribution. Recalling we embed each question, $\mathbf{z}_i = \mathrm{emb}(x_i)$, we run an Adam optimizer to minimize the causal next-word-prediction loss, $\mathrm{argmin}_\theta \sum_{i=1}^n \ell(H(\mathbf{P}_\theta(\mathbf{z}_i), x_i)$, where $\mathbf{P}_\theta(\mathbf{z})$ is the conditioned soft prompt and $H$ is the frozen LLM (post input embedding layer). The sequence embeddings, $\mathrm{emb}(\cdot)$, are computed as the average of token embeddings computed by a small off-the-shelf decoder-only LM. This simple embedding approach is used to both limit the amount of additional computation, but also to ensure the embedding is somewhat lossy in order to make the reconstruction task (i.e. minimizing $\ell(H(\mathbf{P}_\theta(\mathrm{emb}(x_i)), x_i)$ challenging. The simpler SS$_{\mathrm{NSP}}$ method does not use this sequence embedding as it operates with a non-contextual soft prompt.

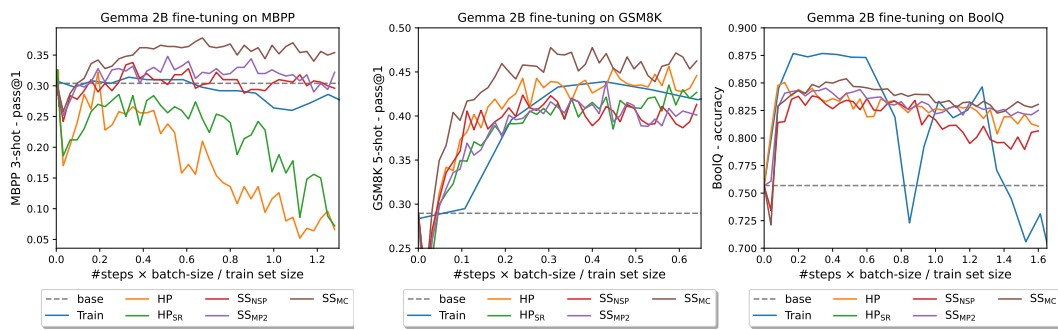

Figure 2: Full fine-tuning curves for the Gemma 2B model using different synthetically generated datasets as well as the non-synthetic training set.

Since we seek an automated hands off approach, we avoid any domain-specific hyperparameter tuning. Specifically, for all SoftSRV variants and all benchmarks, the length of the prompt $t$ is fixed to be 128, the number of training steps was set to 20K, and the learning rate is fixed to $5\mathrm{e}-6$, which we found to be reasonable defaults. The $\mathrm{SS_{MC}}$ method uses MLPs with 3 feed forward layers and 128 hidden dimensions. For the $\mathrm{SS_{MPk}}$ variant, we primarily evaluate with $k = 2$ to limit to the total number of parameters, although a partial exploration for other values of $k$ is presented in Appendix A.4.

**2. Generate Questions.** Once trained, we generate synthetic questions with the SoftSRV model. That is, we pass in the questions from the training dataset, $x_i$, and produce a new sequence $x_i' \sim H(\mathbf{P}_\theta(\mathrm{emb}(x_i)))$ via temperature sampling (with default temp=1). For the $\mathrm{SS_{NSP}}$ variant, no context vector and, thus, no training examples are used during generation. For all SoftSRV methods, no hard prompt template of any kind is used.

For the hard prompt baselines, we generate synthetic questions by querying the same backbone LLM using the relevant domain specific hard prompt template and questions taken from the benchmark training fold to populate the template. We conducted a search over temperature={1,2,4} and found a temperature of 2 to provide a balance of diversity and quality for these hard prompting methods.

Both the SoftSRV and the hard prompt methods use all examples in the training set during this question generation phase. For all methods, we generate 100K questions, repeating example questions from the training fold in a round-robin fashion. We then run a simple filtering, deduplication and subsampling pipeline to arrive at a target fine-tuning dataset size $N_s$. Details of this procedure are provided in Appendix A.1. We use $N_s$=50,000 for MBPP and GSM8K and $N_s$=20,000 for BoolQ.

**3. Generate Answers.** After generating the questions, all methods essentially follow the same procedure to generate answers using an off-the-self LLM. The only difference being, in the case of SoftSRV, we pass the question directly the the LLM without any domain specific prompting to preserve the domain agnostic nature. In the case of hard prompt baselines, we use a domain specific hard prompt template combined with the generated question to query the off-the-shelf LLM for an answer. Once we have full (questions, answer) fine-tuning examples generated, we run a decontamination process to remove any examples that may have been inadvertently leaked to the pretrained LLM, as is standard practice (details provided in Appendix A.2).

**4. Fine-tune and Evaluate Downstream Model.** Finally, for all methods, we use the generated (question, answer) pairs to fine-tune the target Gemma 2B model. We use a batch-size of 16 with sequence length 8192 and with a learning rate with linear warmup from 0 to 1e-6 over 100 steps, followed by a cosine annealing schedule. We evaluate the performance of these fine-tuned models on the test fold of the respective benchmark using the procedure and metric stated in Section 3.1.

### 3.4 GEMMA 2B FINE-TUNE & DOWNSTREAM EVALUATION

Here, we present the performance of Gemma 2B fine-tuned on the generated synthetic datasets. Figures 2 plots the eval metrics for each dataset as a function of the number of fine-tuning steps

Table 1: Downstream task performance of Gemma 2B models fine-tuned on various sources of data. The number of non-synthetic examples used as seed data for the hard prompt and SoftSRV models is reported as $N_r$. The number of synthesized examples post-deduplication, $N_s$, is 50k apart from BoolQ where it is 20k. The base column reports the pre-trained model performance without fine-tuning. The 'train' column corresponds to the model fine-tuned on the non-synthetic examples.

| task | metric | $N_r$ | base | train | HP | $HP_{SR}$ | $SS_{NSP}$ | $SS_{MP2}$ | $SS_{MC}$ |
|---|---|---|---|---|---|---|---|---|---|
| (fixed epoch) | | | | | | | | | |
| MBPP | pass@1 | 384 | 0.304 | 0.314 | 0.254 | 0.250 | 0.334 | 0.324 | 0.348 |
| GSM8K | accuracy | 7,473 | 0.29 | 0.441 | 0.435 | 0.422 | 0.411 | 0.439 | 0.471 |
| BoolQ | accuracy | 9,427 | 0.757 | 0.877 | 0.833 | – | 0.832 | 0.835 | 0.852 |
| (max metric) | | | | | | | | | |
| MBPP | pass@1 | 384 | 0.304 | 0.314 | 0.326 | 0.326 | 0.338 | 0.348 | 0.378 |
| GSM8K | accuracy | 7,473 | 0.29 | 0.441 | 0.456 | 0.435 | 0.424 | 0.439 | 0.478 |
| BoolQ | accuracy | 9,427 | 0.757 | 0.877 | 0.851 | – | 0.838 | 0.845 | 0.854 |

times batch size normalized by training set size (essentially the training epoch modulo an additional constant factor due to sequence packing). These figures show that the model fine-tuned on data generated by the $SS_{MC}$ method generally outperforms the models fine-tuned on the data generated by the other methods. Comparing the hard prompt methods, the HP method outperforms the $HP_{SR}$ method on the GSM8K benchmark. For the MBPP benchmark, HP initially attains a similar performance as that of $HP_{SR}$, but both methods start to degrade as a function of fine-tuning steps. This may be due to a lack of diversity in the generated text given the small set of 384 training example questions for MBPP. We do not report the results of the $HP_{SR}$ method for BoolQ as it appears to struggle to produce reasonable outputs. The repeated self-critiques of $HP_{SR}$ on the lengthy input passages seems to lead it astray from the original intention of the task, producing questions asking for open-ended discussion of a passage rather than targeted true/false questions.

In our soft-prompting setting, two salient questions are what type of parametrization is effective for soft prompts and whether it is essential to have a contextual soft prompt that leverages the context token as opposed to a non-contextual soft prompt. Our empirical evaluations show that the non-contextual soft prompt method, $SS_{NSP}$, admits a lower performance than the other methods thereby indicating the effectiveness of contextual soft prompts. Then, by comparing the performance of the $SS_{MC}$ and $SS_{MP2}$ methods, we observe that the more expressive parametrization of $SS_{MC}$ is generally beneficial.

We also find that the model fine-tuned on the $SS_{MC}$ generated data outperforms the model fine-tuned on the non-synthetic training dataset for MBPP and GSM8K, indicating that, given enough of it, synthetic data can outperform even non-synthetic data. However, the same observation does not hold for BoolQ. The training set curve on BoolQ admits high variance, but it attains a higher accuracy overall. As discussed in Section 3.1, we expect generating questions for the BoolQ dataset to be more difficult stemming both from the fact that the data was generated by search queries thereby covering a wide range of topics, but also from containing much longer sequences extracted from the Wikipedia passages.

In Table 1, we show a comparison of the methods at a fixed epoch and at their max metric value attained over all tested fine-tuning steps. The fixed epoch was chosen to be the step where the model admits the largest fine-tuning performance on the training dataset. This clearly gives an advantage to the model fine-tuned on the training data, but it still provides a fair comparison between the synthetically generated datasets. We again find that the model performance when fine-tuned on the $SS_{MC}$ generated data outperforms the models fine-tuned on the other generated datasets at both the fixed epoch and at the max metric. For the the BoolQ dataset, the HP method admits a performance close to that of the $SS_{MC}$ method for the max metric, but its accuracy for the fixed epoch is considerably lower than that of $SS_{MC}$.

Even though the training datasets from our benchmark domains contain a relatively small number of examples (see column $N_r$ of Table 1 for exact sizes), our methods successfully generate tens of thousands of synthetic examples, thereby showcasing that SoftSRV is well suited for the the data scarce setting.

 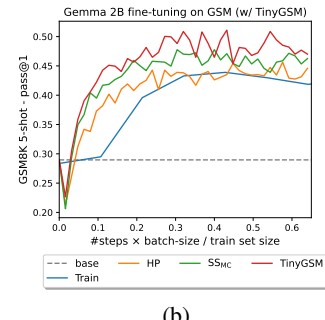

(a)              (b)

Figure 3: In (a) we compare the BoolQ performance of Gemma 2B fine-tuned on data generated by HP and $SS_{NSP}$ as the number of generated examples increases. In (b) for the GSM8K benchmark, we show the Gemma 2B performance after fine-tuning on the $SS_{MC}$ and HP generated datasets against a sample of the same size from the curated TinyGSM dataset.

### 3.5 DATA SCALING

Next, we analyze the effects of varying the number of synthetic examples generated. Specifically, we increase the number of generated examples from 20K, 50K, to 100K, testing both the $SS_{MC}$ and HP approaches for the BoolQ benchmark, given this benchmark appears the most challenging in terms of generating effective synthetic questions. Figure 3a shows that the performance of the model fine-tuned on HP stagnates between 20K and 50K synthetic examples, and only shows improvement when the number of generated examples reaches 100K. In contrast, the model fine-tuned on $SS_{MC}$ steadily improves as the generated dataset size increases. Going from 20K to 100K examples, $SS_{MC}$ performance increases at a 1.8 times faster rate relative to HP with respect to the max metric value. In particular, the HP method with 100K generated examples admits a comparable performance to that of the $SS_{MC}$ method with 50K generated examples.

### 3.6 COMPARISON WITH CURATED HARD PROMPT GENERATED DATASET (TINYGSM)

Here, we compare against a high quality dataset, TinyGSM, within our fine-tuning evaluation framework. TinyGSM was expertly curated by Liu et al. (2023) for GSM8K-PAL, which is a program aided language model (PAL) variant of GSM8K that asks for questions to be answered in the form of Python functions. This has the advantage of enabling verification of the answer in a programmatic fashion. Liu et al. (2023) use GPT-3.5-turbo with hard prompts seeded with training questions from the original GSM8K dataset and from the GSM-IC dataset, which is a dataset crafted to incorporate irrelevant context in order to bolster model robustness (Shi et al., 2023). They use two types of prompts: the first asks to generate both questions and answers while the second requires two calls to the LLM to first generate a question and then an answer. Leveraging the fact that the solutions of the math word problems are written in Python, they then filter out any data that contains code that is not executable by a Python interpreter. They additionally filter out questions that do not contain numbers as this indicates flawed math problems.

In order to evaluate the TinyGSM generated question in our setting, we randomly sample 100K questions from the publicly available TinyGSM dataset, then further subsample down to 50K using the same post-processing pipeline used by all other methods in our comparison. Finally, we generate answers, fine-tune and evaluate in the same fashion as the other hard prompt baselines. Figure 3b shows that a model fine-tuned on the $SS_{MC}$ dataset attains a performance close to that when fine-tuned on the sample from the TinyGSM dataset while the HP method lags behind both. TinyGSM performing closely to $SS_{MC}$ is encouraging given that the TinyGSM dataset is highly curated and tailored specifically for the GSM8K benchmark.

### 3.7 DISTRIBUTION MATCHING ANALYSIS

We next investigate the capacity of the SoftSRV and hard-prompting baseline to generate data samples that can match a target distribution of text. To assess the proximity between the generated and target distributions, we compute Mauve scores as in Pillutla et al. (2021), which can be viewed as a

Table 2: The Mauve similarity scores of synthetic datasets computed with respect to the non-synthetic training and test fold of each dataset. The similarity computed between the train and test fold itself is also included in the final row of the table. Bolding indicates the max Mauve score achieved by a synthetic dataset.

| | Train | | | Test | | |
|---|---|---|---|---|---|---|
| | MBPP | GSM8K | BoolQ | MBPP | GSM8K | BoolQ |
| HP | 0.622 | 0.933 | 0.663 | 0.463 | 0.914 | 0.784 |
| $HP_{SR}$ | 0.327 | 0.870 | – | 0.397 | 0.865 | – |
| $SS_{NSP}$ | **0.776** | 0.862 | 0.519 | **0.781** | 0.839 | 0.575 |
| $SS_{MP2}$ | 0.722 | 0.727 | 0.731 | 0.646 | 0.735 | 0.689 |
| $SS_{MC}$ | 0.604 | **0.993** | **0.997** | 0.477 | **0.991** | **0.995** |
| Train | 1.000 | 1.000 | 1.000 | 0.963 | 0.998 | 0.999 |

scalar summary of the divergence between the textual output of a generative model and a reference distribution. The MAUVE score is able to simultaneously measure both the model's ability to avoid generating text outside the support of the target distribution (Type I error) and the ability to generate text with a large coverage of the target distribution support (Type II error). The method essentially computes a quantized distribution for the generated and target distribution and measures their KL divergence, producing a normalized score between 0 and 1, where 1 indicates the two distributions are maximally similar (for further details see Appendix A.3).

In Table 2, we report the MAUVE scores for hard-prompt and SoftSRV-based synthetic datasets on the MBPP, GSM8K and BoolQ domains, measuring the distance to questions in both the train and test folds. In all cases, we find that a SoftSRV variant can synthesize text that is closer to the training and test dataset distribution than the hard-prompt approaches. This is perhaps unsurprising to see with respect to the train set since, by design, the SoftSRV prompts are trained to optimize the likelihood that the LLM sequentially decodes the text seen in these training examples, but is encouraging to see that the trend holds for the test set as well. Notably, among the SoftSRV variants, the $SS_{MC}$ method achieves the highest score on GSM8K and BoolQ, where it appears the additional flexibility afforded by its parameterization allows a very high-fidelity match to the target distribution. On the other hand, in the case of MBPP, which has relatively simple question distribution (see discussion in Section 3.1), the simplest SoftSRV parameterization ($SS_{NSP}$) attains the largest MAUVE score while the more complex SoftSRV variants produce lower similarity scores, perhaps due to the relatively small MBPP training set (only 384 examples). While the hard-prompt datasets generally have lower MAUVE scores than the SoftSRV counterparts, the HP variant is able to achieve high scores on GSM8K. Notably, in the case of the MBPP dataset, we measure very low similarity scores for $HP_{SR}$; we conjecture this may be due to the several rounds of rewriting, which results in straying further from the original seed question.

Even though the MAUVE score is not a direct indicator of downstream fine-tuning performance (see our experiments in Section 3.4 for downstream fine-tuning comparisons), it is a signal to provide further support for SoftSRV as a high-fidelity approach for text generation across domains.

## 4 RELATED WORK

SoftSRV provides a novel contribution at the intersection of soft-prompting and synthetic text generation for targeted fine-tuning tasks. On each of these individual topics, there is a large and rapidly growing literature, which we touch on only briefly in the section.

### 4.1 SYNTHETIC TEXT FOR LLM TRAINING

As mentioned in the introduction, there is a significant recent body of work demonstrating the effectiveness of using a large language model to generate synthetic training data for a smaller model.

In terms for generating pre-training data, the collection of "Textbooks Are All You Need" white-papers outlines the process of training the Phi series of small LMs, using carefully curated prompt templates and seed data sources, and shows the large boost in quality that synthetic data can provide

(Gunasekar et al., 2023; Li et al., 2023; Abdin et al., 2024). The Cosmopedia project (Ben Allal et al., 2024) conducts a similar study, while also providing detailed and transparent steps as well as open-sourcing the prompts and generated data. Li et al. (2024a) construct a pretraining set almost "from scratch" using a diverse set of hard prompts. To build these prompts, they assume access to taxonomies of fields/sub-fields/disciplines within an area of expertise and build a syllabus which culminates in a series of "key concepts", each of which a large language model is then queried to generate a lesson on. Apart from focusing on pre-training rather than fine-tuning for a specific domain, these works require a non-trivial amount human effort for building and/or curating hard prompts for the generating LLM, which our effort seeks to minimize.

Mukherjee et al. (2023); Mitra et al. (2023) focus on building synthetic data for better instruction tuning. In particular, they start with the FLAN-V2 instruction tuning dataset and ask a LLM to expand on the terse responses in different verbose styles (specified by so-called "system instructions") to introduce variation in presentation and approach. Although, shown to be quite effective across a broad array of reasoning tasks, in our setting we wish to generate data focused on a specific target task, likely requiring us to curate a set of bespoke "system instructions" for each task.

Several works build fine-tuning data for specific domains, such as coding (Haluptzok et al., 2023; Luo et al., 2024) or mathematics (Yu et al., 2024). Although quite successful, these approaches leverage specific qualities of the target domain, for example, using a code interpreter to check correctness of generated code or using the fact that math problems contain numerical quantities that can be masked or manipulated to create variations of the original question.

Finally, in the recent work of Lee et al. (2024), the authors propose an adaptive procedure where a large language model is used to generate targeted fine-tuning data for a small model based on examples that the small model has made mistakes on. The large model is prompted to rewrite variants of these questions using specialized per-domain hard prompts. Extending the general-purpose SoftSRV approach to an adaptive framework is a current area of investigation.

## 4.2 SOFT-PROMPTING & NON-TEXT DATA MODALITIES

As mentioned previously, the use of soft-prompting (or prompt-tuning) already has a significant history outside of targeted synthetic fine-tuning data generation. Primarily known for its use as a parameter efficient fine-tuning method (Lester et al., 2021), soft prompt training has also been used as a framework to learn or compress in-context instructions (Mu et al., 2023). Li et al. (2024b) has proposed training a secondary model to compress in-context instructions into soft prompt that is then prepended to the task hard prompt and is meant to generalize to even new tasks.

While our study has been focused on generating synthetic text, there have been similar efforts in other modalities. For example, the ControlNet approach of Zhang et al. (2023) trains a diffusion model to produce images conditioned on contextual input, for example, image edges or 3D pose skeletons. Similarly, Gao et al. (2024) train a diffusion-based speech model to condition on a "simple speech representation" embedding to guide the generation of new synthetic speech data. Finally, in the case of text-to-image generators, there has been a significant amount of work in solving the "inverse" problem of mapping from an image back to a prompt (either hard prompt or a soft representation), so that one can more predictably generate synthetic images in certain styles (see Mahajan et al. (2024) and many references therein).

## 5 CONCLUSION

In this work, we have established the effectiveness of using contextual soft prompts, via the SoftSRV framework, for generating targeted synthetic fine-tuning data and its applicability across several different domains. We deploy the same SoftSRV pipeline across math, coding, and reasoning tasks, in each case we find SoftSRV is able to generate fine-tuning data that provide good downstream performance and fits well to the target distribution without any per-domain specialization needed. Given these results, there are several natural directions for further research. For example, we can view soft prompts as one particular class of parameter efficient tuning approach that is natural to leverage for data synthethesis, however, other approaches (such as LoRA) may be worth investigating as well. Finally, adapting the choice of context vector in order to generate the most effective synthetic data for improving the downstream model is a current and ongoing line of work.

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

## A   EMPIRICAL EVALUATION ADDITIONAL DETAILS

This appendix provides supplementary details about our empirical evaluations.

### A.1   SYNTHETIC QUESTION POSTPROCESSING

For all methods (hard prompt and SoftSRV based), we query the model to generate 100K sequences. From these 100K, we first filter exact duplicates. From the remainder, we subsample to achieve the target size $N_s$. To encourage a diverse subsample, again for all methods, we cluster the data and select examples from each cluster randomly in round-robin fashion.

Concretely, using the scikit-learn library (Pedregosa et al., 2011), we apply MiniBatch $k$-means to vectorized data, which has been reduced in dimensionality using SVD. For all methods, we set the number of clusters for MiniBatch $k$-means to 700, reduced the dimensionality to 100 features and used sk.TfidfVectorizer for vectorization. Given the $k$-means clustering, we randomly select without replacement one point per cluster until $N_s$ questions are chosen.

### A.2   DECONTAMINATION PROCESS

Even though the test set was never used in our synthetic data generation pipeline, the frozen LLM models that are leveraged to generate questions and answers might have been exposed to the test set during their pretraining phase. Thus, for all methods, we decontaminate the generated sequences against the respective benchmark's test set by removing any n-gram matches where $n = 13$ as is common practice (Brown et al., 2020). Prior to calculating the matches, we eliminate all punctuation and numerical characters. We found that the contamination of the generated sequences to the test set is minimal with less than $0.1\%$ for GSM8K and MBPP and around $1\%$ for BoolQ.

### A.3   MAUVE SCORE COMPUTATION

We let $\mathcal{G}$ and $\mathcal{D}$ denote the generated and target/reference distributions, respectively, and compute the MAUVE scores for each synthetic dataset as follows. Using an embedding model, a vector representation is computed for each sequence in the synthetic and reference sets. These embeddings are then projected into a discrete set using $k$-means clustering, and a divergence curve is traced between the cluster distributions of $\mathcal{G}$ and $\mathcal{D}$, see Equation 1 in Pillutla et al. (2021). The MAUVE has value between 0 and 1 and corresponds to the area under the divergence curve, where higher scores are indicative of a closer match between $\mathcal{G}$ and $\mathcal{D}$. The same small LM serving as the context embedder for SoftSRV in Subsection 3.3 is used to compute per-token representations, which is then averaged to produce the sequence-level embedding. We use $k = 32$ clusters for all domains as we found that $k = 16$ or $k = 64$ yields similar qualitative results.

### A.4   MIXTURE OF PROMPTS WITH VARIOUS VALUES OF $k$

Here, we conduct a exploration to measure the effect of changing the number of basis soft prompt matrices, $k$, of the $\text{SS}_{\text{MPk}}$ method.

Figure 4 shows the comparison in performance on various benchmarks across different values of $k$. We see an inherent trade-off as we increase $k$, which increases the capacity of the contextual soft-prompt but also then require training more parameters. For MBPP we see that the value $k = 4$ appears to be optimal, while for GSM8K and BOOLQ we see performance peaks at $k = 2$. We fix $k = 2$ throughout the evaluations in the main paper as a good general choice across different tasks.

### A.5   DIVERSIFICATION OF THE HP METHOD

To arrive at the the HP method presented in the main body of the paper, we conducted multiple iterations of prompt engineering and template refinements. In particular, we found that asking the model to generate "10 different questions" per example question and using a higher decoding temperature was critical. We demonstrate this effect on the GSM8K benchmark in Figure 5 which compares the performance of a model fine-tuned on data generated by HP method to that of the model fine-tuned on its undiversified counterpart where the question template asks to generate one question per

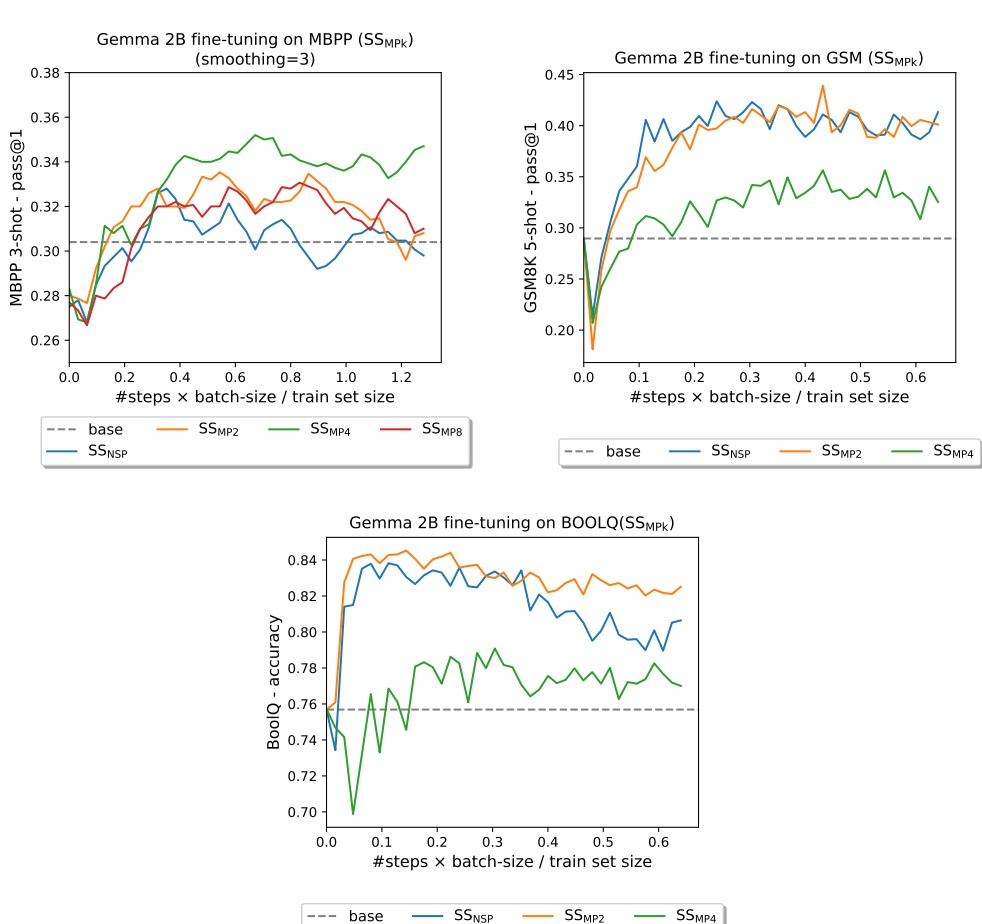

Figure 4: Comparison of $\mathrm{SS_{MPk}}$ variants for different values of $k$. $\mathrm{SS_{NSP}}$ corresponds to $k = 1$.

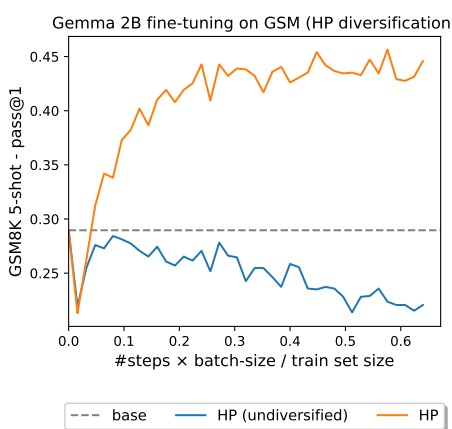

Figure 5: Performance of HP method with and without diversification.

given example question and the default decoding temperature is used. We found similar results for the other datasets. Despite the relative small change in the settings, the difference in model performance is significant, demonstrating some of the idiosyncratic nature of hard-prompting approaches. We provide the template for the undiversified HP in Appendix B.

## B   HARD PROMPT TEMPLATES

Below, we report the templates used for hard-prompting baselines for each benchmark dataset. Figure 6 provides the question templates while Figure 7 shows the answer templates. Figure 8 reports the critique and refine templates for $HP_{SR}$. Figure 9 provides the template for the undiversified HP method, described in Appendix A.5, for the GSM8K benchmark .

**MBPP Question Template:**

```
Consider the following python question:
              [insert example question]
Now generate 10 different questions that require writing a Python
function similar to the example above.  Make sure each question is
different and sufficiently rephrased.  Please make sure you generate
questions, and not answers.  Please make sure each question you
generate has a well-defined answer.
Question 1:
```

**GSM8K Question Template:**

```
Consider the following grade-school math problem:
              [insert example question]
Now generate 10 different questions that require solving a
grade-school math problem similar to the example above.  Make sure
each question is different and sufficiently rephrased.  Please make
sure you generate questions, and not answers.  Please make sure each
question you generate has a well-defined answer.
Question 1:
```

**BoolQ Question Template:**

```
Consider the following passage and question:
              [insert example question]
Now generate 10 different passages and questions similar to the
example above.  Please make sure each question you generate has a
boolean answer that can be answered by the passage.  Make sure each
passage and question is different and sufficiently rephrased.  Please
make sure you generate passages and questions, and not answers.
Passage and Question 1:
```

Figure 6: Question template for MBPP, GSM8K and BoolQ benchmarks for the HP method.

**MBPP Answer Template:**

```
Please answer the following python question:
              [insert example question]
Please generate your answer as a Python function.  The docstring of
the function should contain the above question as-is, without any
modification.  Please make sure that your function is valid Python
code that compiles.  Please try your best to correctly answer the
question.
Answer:
```

**GSM8K Answer Template:**

```
Please answer the following question that tests reasoning:
              [insert example question]
Answer:
```

**BoolQ Answer Template:**

```
Please answer the following question based on the passage.  Your
answer should be either True or False.  Do not provide any other
justification.
              [insert example question]
Answer:
```

Figure 7: Answer template for MBPP, GSM8K and BoolQ benchmarks for the HP and $\text{HP}_{\text{SR}}$ method.

**Critique Template:**

```
Please provide actionable feedback on the clarity, difficulty, and
originality of the following {Python question, grade school math
problem, passage/question problem}:
                    [insert question]
```

**MBPP Refine Template:**

```
Read the following Python question and the critique, and write a new
Python question based on the critique:
Question:
                    [insert question]
Critique:
                    [insert critique]
If the critique is strongly positive, say 'Stop'.  Otherwise, write
a new Python question in a single sentence starting with 'Write a
Python function' based on the critique.  Do not ask for docstring or
test cases.
```

**GSM8K Refine Template:**

```
Read the following grade-school math problem and the critique, and
write a new grade-school math problem based on the critique:
Question:
                    [insert question]
Critique:
                    [insert critique]
If the critique is strongly positive, say 'Stop'.  Otherwise, write
a new grade-school math problem based on the critique.  Write the
question only, do not include the answer.
```

**BoolQ Refine Template:**

```
Read the following passage/question problem and the critique, and
write a new passage/question problem based on the critique:
Question:
                    [insert question/passage]
Critique:
                    [insert critique]
If the critique is strongly positive, say 'Stop'.  Otherwise, write
a new passage/question problem based on the critique.  Write the
passage and question only, do not include the answer.
```

Figure 8: Refine template for MBPP, GSM8K and BoolQ benchmarks for the $HP_{SR}$ method.

**GSM8K Question Template for undiversified** HP**:**

```
 Please generate a question that requires solving a grade-school math
problem.  Here is an example of such a question:
                [insert example question]
Now generate a new question.  Please make sure your question is not
too similar to the example above.  Please make sure you generate
a question, and not an answer.  Please make sure the question you
generate has a well-defined answer.
```

Figure 9: Question template for the undiversified HP method for the GSM8K benchmark.

