# OpenReview forum: "No more hard-prompts: SoftSRV prompting for synthetic data generation"
_ICLR.cc/2025/Conference — Submitted to ICLR 2025_

### Official Review · Reviewer_5RCw · 2024-10-17

**Soundness:** 2
**Presentation:** 3
**Contribution:** 2
**Rating:** 3
**Confidence:** 4

**Summary:**

This paper proposes SoftSRV, which trains and adapts soft prompt in synthetic data generation, while previous works mainly generate pseudo data by manually hard prompts. Empirical results on diverse tasks show that the models trained by SoftSRV-generated data performs better than those trained on data generated by baseline hard-prompting approaches. Besides, softSRV always generates data with better matches the target distribution.

**Strengths:**

* This paper well written and the analysis of distribution matching is clear.
* The experimental results compared with hard prompt baselines are promising.

**Weaknesses:**

* SoftSRV is only applied on synthetic data generation, actually the application can be more diverse and broader. What about the performance on some specific downstream tasks if we can train soft prompts on these tasks? Will it perform better than other soft prompts like Prefix-tuning[1], RLPrompt[2], P-tuning[3,4]?
* The paper mentioned SoftSRV is different from previous soft prompt methods, rather than prepending parameters, it instead uses the soft prompt alone as input context to the LLM. But it does not give some comparisons between SoftRSV and these soft prompt methods. What if those soft prompt methods be used in synthetic data generation?
* To some extent, this paper only adapts soft prompts in pseudo data generation, rather than propose a brand new, novel method. The contribution might not meet the level of substantial novelty expected at a conference like ICLR. So if SoftSRV performs better than other soft prompt baselines on other downstream tasks instead of only data generation, I'll consider increasing my score.

[1] Li, Xiang Lisa, and Percy Liang. "Prefix-tuning: Optimizing continuous prompts for generation." arXiv preprint arXiv:2101.00190 (2021).

[2] Deng, Mingkai, et al. "Rlprompt: Optimizing discrete text prompts with reinforcement learning." arXiv preprint arXiv:2205.12548 (2022).

[3] Liu, Xiao, et al. "GPT understands, too." AI Open (2023).

[4] Liu, Xiao, et al. "P-tuning v2: Prompt tuning can be comparable to fine-tuning universally across scales and tasks." arXiv preprint arXiv:2110.07602 (2021).

**Questions:**

* What about the performance on some specific downstream tasks of SoftSRV compared with other soft prompt methods?
* This paper only compares SoftSRV with hard prompt baselines on synthetic data generation. More comparisons with previous soft prompt methods applied in this data generation area are necessary.

---

> ### Author Response · Authors · 2024-11-25
> **Can standard soft-prompt methods be used for synthetic data generation?**
>
> We want to stress that standard soft-prompt methods (e.g. prefix-tuning) are not directly applicable to synthetic data generation. Concretely, consider a standard soft-prompt method trained on a question/answer task, the standard soft-prompt methods will try to generate the answer given the question as opposed to generating a similar question on the same topic.  In order to generate a similar question on topic using standard soft-prompt methods, one would, for example, need to create a dataset with pairs of similar questions and then train a soft-prompt method on this new dataset. Building a dataset of paired examples is challenging due to a potential scarcity of similar examples and difficulty in defining a notion of similarity.
>
> Our goal is to develop a synthetic data generation framework and targeting standard soft-prompt setting is beyond the scope of this paper.

---

### Official Review · Reviewer_pGWo · 2024-10-30

**Soundness:** 2
**Presentation:** 3
**Contribution:** 2
**Rating:** 5
**Confidence:** 4

**Summary:**

In this paper, authors want to introduce a data synthetic generation method by using a soft-prompt based framework. It aims to train a parameterized soft-prompt by using data-driven loss minimization, and thus synthesize sequences to satisfy the target distribution $D$. Specifically, they found that the parameterized families of soft prompt can be conditioned on an input context and can fit the target distribution. Experimental results indicate that the proposed method can improve the fine-tuned model to achieve better performance based on the synthetic data.

**Strengths:**

**Strengths**

1. This paper introduces how to train soft prompts to build a data synthetic generation to fit target distribution.
2. Experimental results demonstrate the effectiveness of the proposed method can fit the target distribution.

**Weaknesses:**

**Weaknesses**

1. To build such a data synthetic generation, the proposed method first needs to obtain a lot of samples from the target distributions to train the soft prompt. So, which is advantage to use the proposed method as directly use hard prompt does not require any target samples for training?
2. The generalization of the proposed method is also a concern. It seems if we want to use a different LLM (e.g., LLama-70B) for data generation, we also need to train a corresponding model. Therefore, the proposed method (i.e., the trained soft prompt) cannot be adapted to different LLMs, while hard prompt does not have such a concern.
3. Experiments on more LLMs are required. In this paper, authors only use Gemma-2B as a backbone network.
4. In authors' setting, training SoftSRV also require some training (e.g., 20K steps). So what happened if our downstream tasks do not have enough data for training?
5. In this paper, authors only choose three standard datasets (i.e., Code, Math and Reasoning) for generation. Do you try some other open-domain scenarios, like some datasets which are not in question-answer format (e.g., Chat)?

**Questions:**

Please see my comments on Weaknesses. I am willing to increase my score if authors can address my concerns.

---

> ### Author Response · Authors · 2024-11-25
>
> * Question: "To build such a data synthetic generation, the proposed method first needs to obtain a lot of samples from the target distributions to train the soft prompt. So, which is advantage to use the proposed method as directly use hard prompt does not require any target samples for training?"
>
> The hard prompt methods do require target samples to populate the templates, that is the templates use example questions that are taken directly from the training set. See Section 3.2 and Appendix B.  To train SoftSRV, we use the same amount of training data as hard prompting methods. From mere 100-1000s of examples, we create 10x as many synthetic examples.
>
> * Question: "The generalization of the proposed method is also a concern. It seems if we want to use a different LLM (e.g., LLama-70B) for data generation, we also need to train a corresponding model. Therefore, the proposed method (i.e., the trained soft prompt) cannot be adapted to different LLMs, while hard prompt does not have such a concern."
>
> This is true, but also would be true of any training based approach.  Even for hard-prompts, one would potentially need to refine the hard-prompt for each model type to match the idiosyncrasies of each model. This can be due to, for example, different instruction tuning procedures for each model.
>
> * Question: "In authors' setting, training SoftSRV also require some training (e.g., 20K steps). So what happened if our downstream tasks do not have enough data for training?"
>
> We find that 100-1000's of examples is sufficient to train SoftSRV.  Note that the hard-prompt baselines also use this data to seed the hard prompts templates.
>
>
> * Question: "Experiments on more LLMs are required. ... Do you try some other open-domain scenarios, like some datasets which are not in question-answer format (e.g., Chat)?"
>
>  We agree that trying open-domain scenarios and testing more LLMs is an important future direction, which we plan on pursuing.

---

> > ### Comment · Reviewer_pGWo · 2024-11-28
> >
> > Thanks for your answers. After reading the response, I will maintain my score as authors also admit the problem of generalization in the proposed method.

---

### Official Review · Reviewer_MiC5 · 2024-11-03

**Soundness:** 3
**Presentation:** 2
**Contribution:** 2
**Rating:** 5
**Confidence:** 4

**Summary:**

The paper introduces SoftSRV, a prompt-tuning framework designed to synthesize data with frozen LLMs. Unlike standard prefix-tuning prepending a soft prompt to an existing hard prompt, SoftSRV uses only the soft prompt as input context for the LLM. The paper presents three structures of SoftSRV: $SS_{NSP}, SS_{MPk}, SS_{MC}$. The authors then fine-tune SoftSRV on coding, mathematics, and reasoning domains to synthesize data. Experiments show that fine-tuning Gemma-2B on SoftSRV synthesized data outperforms zero-shot synthesized data on those domains.

**Strengths:**

- The target problem is practical and useful.
- The proposed method is simple and intuitive.
- Experiments show the promise of the proposed method and the improvements are also very intuitive.

**Weaknesses:**

- The method shows limited novelty. It is evident that generating data using a PEFT fine-tuned model on specific domains improves domain alignment, outperforming zero-shot data generation. A model fine-tuned on ~20K samples is naturally better aligned with these domains than zero-shot, which relies on only 0 sample.
- Given this, the approach appears straightforward. Rather than fine-tuning a model for data generation, why not simply PEFT fine-tune the model directly on these tasks? As shown in Table 1, the "train" column demonstrates superior performance compared to the "HP" columns.
- Testing with out-of-distribution (OOD) data instead of MBPP, GSM8K, and BoolQ could further validate the method’s robustness.
- The paper evaluates only one LM; assessing additional LMs could strengthen its claims.
- Current comparisons are unfair. More baselines that incorporate training-based data generation methods are needed.

**Questions:**

See weaknesses.

---

> ### Author Response · Authors · 2024-11-25
>
> * Question: "The method shows limited novelty.... A model fine-tuned on ~20K samples is naturally better aligned with these domains than zero-shot"
>
> We are not comparing to the zero-shot setting. All methods, including the hard-prompt baselines, are fully fine-tuned on the same number of synthetic data (see Figure 2).  The comparison is between fine-tuning on synthetic data generated by hard prompting methods versus by SoftSRV methods.
>
> Perhaps the reviewer’s question is about question generation? At question generation time, both hard prompting methods and SoftSRV use data from the benchmark's training fold. In particular, the hard-prompt seeds the prompt template with examples from the training fold.  Please see section 3.3 for more details.
>
>
> * Question: "Given this, the approach appears straightforward. Rather than fine-tuning a model for data generation, why not simply PEFT fine-tune the model directly on these tasks? As shown in Table 1, the "train" column demonstrates superior performance compared to the "HP" columns."
>
> In Table 1, we are fully fine-tuning Gemma 2B where each column indicates the data we are using (e.g. original training data, synthetic data generated by HP, etc). We find that fine-tuning using the SS_MC generated data performs better than fully fine-tuning on the original training data for all datasets except BOOLQ. See discussion in line 357-363.  In general, we expect that PEFT Gemma on the original training data would perform worse compared to fully finetuning Gemma on the original training data.  Note also that HP columns are the competitor baseline methods.
>
> Perhaps the reviewer is asking about PEFT the large model to use on the downstream task directly? This is a different experiment and we are imagining a (typical) scenario where we want to use a small downstream model, for example, for serving efficiency.
>
> Perhaps the reviewer is asking about using PEFT for synthetic data generation? Please see discussion with Reviewer 5RCw.
>
>
> * Question: "Current comparisons are unfair. More baselines that incorporate training-based data generation methods are needed."
>
> The hard prompting baselines do use the training data and we are not aware of other published baselines that are conceptually different than the ones we already tested.
>
>
>
> * Question:  "Testing with out-of-distribution (OOD) data instead of MBPP, GSM8K, and BoolQ could further validate the method’s robustness. The paper evaluates only one LM; assessing additional LMs could strengthen its claims."
>
> We agree that evaluating on out-of-distribution domains and testing more LLMs is an important future direction, which we plan on pursuing.

---

> > ### Comment · Reviewer_MiC5 · 2024-11-26
> > **Official comment by reviewer MiC5**
> >
> > Thank you for clarifying my first two questions, they addressed my points there. My main concern is about the generalization of this work. Therefore, I tend to recommend rejection. I increased the soundness and overall rating.

---

### Official Review · Reviewer_NGUv · 2024-11-03

**Soundness:** 3
**Presentation:** 4
**Contribution:** 3
**Rating:** 6
**Confidence:** 3

**Summary:**

This paper suggests an alternative to hand engineering/crafting prompts for designing synthetic data, which is based on soft prompting to learn soft tokens or embedding strategies that minimize NLL on the small amount of human data available, and then generating conditioned on those soft tokens.

**Strengths:**

This paper addresses a very real problem (the generation of synthetic data) in a novel way.  Their approach is fairly rigorous and the experimental sections contain lots of information; for instance, they report on three variations of their idea (SS_{NSP}, SS_{MPK}, SS_{MC}) and compare performance on a diverse number of metrics (including both downstream task performance after finetuning on the dataset and the human baseline coverage.

The experiments demonstrate quite convincingly that the approach does better than hand-engineering prompts to generate this synthetic data **when enough human examples are available to tune the prompts**.  Furthermore, it is clear that this is a more "sustainable" approach, as it is not feasible to generate hand-engineered prompts for every domain if you have many domains you are perhaps interested in.

**Weaknesses:**

I could be wrong, but it seemed to me like they only applied this in domains where it was possible to learn an entire neural network to maximize the likelihood of the "real" data without overfitting, which I imagine is only the case when you have a lot of human data.  To me, this seems like the least likely instance where you'd need to do synthetic generation.  Thus, while I'm convinced the application is real and the empirical results are valid, I would imagine (but would be happy to be proven wrong) that the intersection of problems where you **could** apply this approach and problems where you'd **need** to apply this approach is fairly limited.


Furthermore, I see no novel algorithms or mathematical foundations in this paper.  It is a very straightforward application of an approach developed by Lester et al and Li and Liang (who really should be cited), just to this novel task.  And, since, as I pointed out earlier, there may not be many use cases for this task, it may be that the only reason no one has done this yet is that it isn't a very useful thing to do in practice.  However, again, I'd be happy to be proven wrong.

**Questions:**

What is the minimum amount of human data you need to make this approach not overfit?

How would you generalize this approach to when you only have `k` examples of human data?  (This is asking a bit much, but I'm curious if you've thought about it)
?
Do you see your paper as having novel mathematical contributions, and, if so, what are they?

---

> ### Author Response · Authors · 2024-11-25
>
> * Question: “only [useful] case when you have a lot of human data”
>
> The parametrized soft-prompts have a relatively small number of parameters, compared to the frozen LM, and can be trained with a small amount of data (even just hundreds of examples). Specifically,  MBPP only has 384 training examples while GSM8K and BOOLQ have ~7K and ~9K training examples respectively. Once the parameterized soft-prompt is trained, we then greatly expand the fine-tuning set generating tens of thousands of synthetic examples. Thus, we view SoftSRV as very useful in the data scarce setting.
>
>
> * Question: “application of Lester et al and Li and Liang (who really should be cited)”
>
> Indeed we meant to cite Li and Liang, this was an oversight that is now remedied.  However, we do not view it as a straightforward application, given that we introduce the idea of parameterized *contextual* soft-prompts that allows us to generate a more diverse and representative set of synthetic data. Additionally, unlike Lester et al and Li and Liang, we are not pre-appending our soft-prompts to the input, but are effectively replacing the input with the soft-prompts, see Figure 1.  Lastly, the standard soft-prompt methods are not directly applicable to synthetic data generation.  See discussion with Reviewer 5RCw.
>
>
> * Question: “minimum amount of human data”
>
> We find, in our experimental setting, hundreds to a few thousand examples were sufficient, even for the most complex soft prompt family SS_MC. As mentioned above, MBPP only has 384 training examples.
>
>
> * Question: "Do you see your paper as having novel mathematical contributions, and, if so, what are they?"
>
> Theoretically, it would be interesting to show that generating a sample from the SoftSRV framework is close to drawing a sample from the target distribution.  Previous work has shown the soft-prompt guide the LLM towards the target task, but arguing the generated data distribution is close to the target distribution is a more difficult task.

---

> > ### Comment · Reviewer_NGUv · 2024-11-25
> > **Thank you for your comment**
> >
> > I have adjusted my score, as my concerns were answered somewhat effectively.

---

### Official Review · Reviewer_zpvo · 2024-11-04

**Soundness:** 3
**Presentation:** 3
**Contribution:** 3
**Rating:** 5
**Confidence:** 3

**Summary:**

The paper introduces SoftSRV, a novel framework that uses soft prompts to generate synthetic training data using frozen large language models (LLMs). Rather than relying on manually crafted hard prompts, SoftSRV learns parameterized "contextual" soft prompts through data-driven optimization. The authors evaluate three variants of their approach (SSNSP, SSMPk, SSMC) across different domains (coding, math, reasoning) and show superior performance compared to hard-prompting baselines when using the generated data to fine-tune smaller models.

**Strengths:**

- Requires minimal human intervention
- Introduces contextual conditioning for better distribution matching
- Outperforms hard-prompting baselines across multiple domains
- Shows better distribution matching (MAUVE scores)
- Supports different soft prompt architectures
- Demonstrates practical alternative to manual prompt engineering

**Weaknesses:**

1. Domain-Agnostic Parameters: Assumes fixed hyperparameters (prompt length=128, 20K training steps, learning rate=5e-6) work across domains

2. Sufficiency of Context Vector: Assumes the context vector derived from an example sequence captures enough information to generate meaningful variations

3. Small Training Sample Sensitivity: For datasets with small training sets (like MBPP with only 384 examples), more complex SoftSRV variants perform worse than simpler ones, suggesting the approach may be sensitive to training sample size.

4. Task Complexity Impact: The approach appears less effective for more complex tasks like BoolQ that require generating longer passages and more diverse content. The authors note this is "perhaps the most difficult task to generate synthetic data for."

5. No Direct Performance Indicator: The authors note that the MAUVE similarity score they use to measure distribution matching is "not a direct indicator of downstream fine-tuning performance," suggesting a lack of clear metrics to predict effectiveness.

6. Problem Setup Limitations:
- Assumes fixed maximum sequence length m (Section 2, pg 2)
- Restricts to scenarios where input and output sequences have equal length

7. Methodological Concerns:
- Relies heavily on a "lossy" sequence embedder without strong justification
- No clear guidance on how to select the degree of "lossiness"

8. Validation Gaps:
- Initial results focused on only three domains (coding, math, reasoning)
- No clear guidelines for choosing between different variants (SSNSP, SSMPk, SSMC)

9. Heavy reliance on MAUVE score which is acknowledged to not directly indicate downstream performance

10. Comparison Scope:
- Primarily compares against hard-prompting baselines
- Limited comparison with other synthetic data generation approaches
- No comparison with other parameter-efficient tuning methods

**Questions:**

1. How sensitive is the approach to the quality and diversity of the initial sample data from the target distribution?
2. What is the minimal sample size needed for effective training across different domains?
3. How does the choice of sequence embedder affect performance?
4. How well does the approach handle very specific or niche domains not well-represented in the LLM's training data?
5. Why choose MLPs for the SSMC variant?
6. How was the number of basis prompts (k=2) chosen for SSMPk? What's the tradeoff between k and performance?
7. How robust is the approach to different random seeds and initialization?
9. How does it compare to other synthetic data generation approaches beyond hard prompting?

---

> ### Author Response · Authors · 2024-11-26
>
> * Question: "Small Training Sample Sensitivity: For datasets with small training sets (like MBPP with only 384 examples), more complex SoftSRV variants perform worse than simpler ones, suggesting the approach may be sensitive to training sample size."
>
> The results in Figure 2 demonstrate that the most expressive SoftSRV variant, SS_MC, consistently exhibits superior finetuning performance, irrespective of the training set size. While Section 3.7 highlights that certain SoftSRV variants achieve improved MAUVE scores compared to SS_MC, this analysis is secondary to our primary objective of enhancing finetuning performance.
>
> The goal of Section 3.7 is that SoftSRV methods result in a synthetic data that is more aligned with the underlying true distributions compared to hard prompted methods (HP, HP_SR) since SoftSRV directly optimizes a data-driven objective guiding the pre-trained model towards the target distribution.
>
> * Question: "No Direct Performance Indicator: The authors note that the MAUVE similarity score they use to measure distribution matching is "not a direct indicator of downstream fine-tuning performance," suggesting a lack of clear metrics to predict effectiveness."
>
> The writing in that paragraph is convoluted and it has misled the reviewer. We will make sure to revise it. Here, we wanted to point out that the MAUVE should only be used as a tool for a secondary analysis since it is not a direct indicator of downstream performance.  In the previous sections, we do show that SoftSRV admits direct downstream performance improvements on the test set – please see Figure 2 and Table 1.
>
> * Question: "Problem Setup Limitations: Assumes fixed maximum sequence length m (Section 2, pg 2). Restricts to scenarios where input and output sequences have equal length"
>
> All LLM have some max length in practice.  Input and output sequences do not need to be the same and in Section 2, we assume them to have equal length only for notional simplicity, without loss of generality.  Note, in our experiments, input and output sequences are not necessarily the same.
>
>
> * Question: "Comparison Scope: Primarily compares against hard-prompting baselines. Limited comparison with other synthetic data generation approaches. No comparison with other parameter-efficient tuning methods"
>
> All prior synthetic data generation approaches that have shown promising results are based on hard prompting. We would be grateful for the reviewer to point us to other promising approaches in this setting.
>
> This paper focuses on developing a new data-driven framework for generating synthetic data. While SoftSRV could potentially be applied to parameter-efficient tuning methods, exploring that application is outside the scope of our current research. Note that parameter-efficient tuning methods cannot be directly applied to targeted data generation. See discussion with Reviewer 5RCw.
>
>
> * Question: "Validation Gaps: Initial results focused on only three domains (coding, math, reasoning). No clear guidelines for choosing between different variants (SSNSP, SSMPk, SSMC)."
>
> The SS_MC method outperforms the other SoftSRV methods on all three domains and hence our suggestions is to use this method.  We will clarify this in the paper.
>
> * Question: Reviewer asks a series of questions about SoftSRV hyperparameters & empirical setup (prompt length, training steps, learning rate, quality and diversity of the initial sample, minimal sample size, choice of sequence embedder, niche domains, why use MLPs, basis prompts for SSMPk, random seeds, and initializations).
>
>  While tuning hyperparameters for SoftSRV (e.g. choice of embedder, SoftSRV parametrizations, etc.) or varying the underlying setting (e.g. training set size, random seeds/initiatilizations, etc) could yield interesting insights and further performance improvements, we of course have limited time and compute for the study and find our methods already outperform hard prompt baselines even when fixing these hyperparameters and experimental setup a priori.
>
> * Question: "Sufficiency of Context Vector: Assumes the context vector derived from an example sequence captures enough information to generate meaningful variations"
>
> We showed empirically that the context vector contains sufficient information to beat hard-prompting baseline methods. It would indeed be interesting to conduct a theoretical analysis to further understand the sufficiency of context vectors.
>
>
> * Question: "Task Complexity Impact: The approach appears less effective for more complex tasks like BoolQ that require generating longer passages and more diverse content. The authors note this is "perhaps the most difficult task to generate synthetic data for."
>
> Yes, BoolQ is the hardest task among the three, but nevertheless we still find that SS_MC admits the best performance.

---

> > ### Comment · Reviewer_zpvo · 2024-11-27
> > **Thank you for your response**
> >
> > Thank you for your responses. However, I would like to keep my score.

---

### Meta-Review · Area_Chair_Zh2D · 2024-12-20

**Metareview:**

This paper proposes a simple approach for synthetic data generation which learns soft prompts on a small amount of data that can subsequently be used for data generation. On the positive side the approach is simple, and coupled with some empirical results that show that it may be promising. On the negative side, it is unclear how practical the approach is, whether it would be generlizable. and whether the evaluation of the method (e.g, using MAUVE) is reliable.

**Additional Comments On Reviewer Discussion:**

Many reviewers pointed out that this approach is not too novel (given the extensive line of work on prompt tuning) and would be unlikely to generalize to other tasks/datasets. All reviewers except for one chose to maintain their score after engaging with the authors.

---

### Decision · Program_Chairs · 2025-01-22

Reject